# Impact of Cabozantinib Exposure on Proteinuria and Muscle Toxicity in Patients with Unresectable Hepatocellular Carcinoma

**DOI:** 10.3390/ph15121460

**Published:** 2022-11-25

**Authors:** Hironao Okubo, Hitoshi Ando, Yusuke Takasaki, Eisuke Nakadera, Yuka Fukuo, Shuichiro Shiina, Kenichi Ikejima

**Affiliations:** 1Department of Gastroenterology, Juntendo University Nerima Hospital, Tokyo 177-8521, Japan; 2Department of Cellular and Molecular Function Analysis, Kanazawa University Graduate School of Medical Sciences, Kanazawa 920-8640, Japan; 3Department of Gastroenterology, Juntendo University School of Medicine, Tokyo 113-0034, Japan

**Keywords:** cabozantinib, pharmacokinetic analysis, muscle enzyme, hepatocellular carcinoma

## Abstract

This prospective study investigated the impact of cabozantinib exposure on proteinuria and muscle toxicity, in a cohort of 14 Japanese patients with unresectable hepatocellular carcinoma (uHCC). We measured the trough concentration of cabozantinib (C_trough_) weekly for 6 weeks after starting treatment. Although the initial dose was less than 60 mg in most cases, dose interruption occurred in 79%, primarily because of proteinuria and/or malaise. The median and coefficient of variation of maximum C_trough_ at 7–42 d were 929.0 ng/mL and 59.2%, respectively. The urinary protein-to-creatinine ratio (UPCR), serum creatine kinase, and serum aldolase values were all significantly elevated following treatment. Moreover, maximum changes in serum creatine kinase and aldolase were significantly associated with maximum C_trough_ (*r* = 0.736, *p* < 0.01; *r* = 0.798, *p* < 0.001; respectively). Receiver operating characteristic (ROC) curve analysis showed that changes in serum creatine kinase ≥70.5 U/L and aldolase ≥6.1 U/L from baseline relatively accurately predicted inclusion in the high-maximum C_trough_ (≥929.0 ng/mL) group, with an area under the ROC of 0.929 and 0.833, respectively. Measurement of serum creatine kinase and aldolase may increase the clinical usefulness of cabozantinib treatment for uHCC and help alleviate difficulties with dose adjustments.

## 1. Introduction

Hepatocellular carcinoma (HCC) is the fourth leading cause of cancer death worldwide [1]. Cabozantinib is an oral tyrosine kinase inhibitor (TKI) that inhibits vascular endothelial growth factor receptor (VEGFR)-1-3, hepatocyte growth factor receptor/MET, and growth arrest-specific 6 receptors [2]. Cabozantinib has been administered following the use of other systemic therapies for unresectable HCC (uHCC), and the initial recommended dose was set at 60 mg [3]. However, the rates of cabozantinib dose reduction because of several treatment-related adverse events (AEs) were 62% in the phase III CELESTIAL trial [3] and 91% in the Japanese phase II trial [4]. Moreover, real-world data from other cohorts on the use of cabozantinib for uHCC showed starting doses of 40 or 20 mg [5,6]. Recent real-world data revealed that the common AEs were malaise (68%), diarrhea (54%), anorexia (46%), and hand and foot syndrome reaction (44%) [7]. Although proteinuria was observed in less than 5% of subjects in the CELESTIAL trial and in those from real-world data with cabozantinib [3,7], it occurred in 21% of patients in the Japanese phase II trial [4]. In the CELESTIAL trial, AEs such as hand and foot syndrome reaction, diarrhea, and hypertension during treatment were reported to be associated with drug exposure [8]. However, no studies have reported the exposure–toxicity relationship of cabozantinib for uHCC patients in clinical practice, which causes difficulties with dose setting and modification. Therefore, we aimed to clarify the association between cabozantinib exposure and the development of AEs, including proteinuria and muscle toxicity, in uHCC patients.

## 2. Results

### 2.1. Patient Features 

Fourteen Japanese patients (1 female and 13 males), including one dialysis patient, with median age of 80 years were enrolled in the study. Baseline features of patients are shown in Table 1. The treatment line was 2nd in six patients, 3rd in seven patients, and 5th in one patient. The modified albumin-bilirubin grade was 1 in two patients, 2a in eight patients, and 2b in four patients. Six of 14 patients (43%) had past history with atezolizumab plus bevacizumab, and 13 of 14 patients (93%) had a history using lenvatinib. Regarding the initial dose of cabozantinib, one patient received 60 mg, eight patients received 40 mg, four patients received 20 mg, and the dialysis patient received alternate-day dosing of 20 mg. Median (range) baseline UPCR, excluding the dialysis patient, was 0.202 (0.06–1.98) g/gCr. Median (range) baseline serum levels of creatine kinase (normal, males: 57–240 U/L, females: 47–200 U/L) was 80 (28–154) U/L, while median (range) baseline serum aldolase (normal, 2.1–6.1 U/L) was 5.25 (3.1–6.6) U/L.

### 2.2. Therapeutic Details and Safety

With a median follow-up of 141 days, as shown in Table 2, the median duration of cabozantinib exposure was 70 days. There was adequate compliance with medication with a median relative dose intensity of 33.3% at 2 weeks and 30.6% at 4 weeks. Dose reduction because of any AE occurred in 100% of patients, and median duration of the first dose reduction was 14 days. Additionally, dose interruption because of any AE occurred in 79% (11/14) of patients, and the first median duration of dose interruption was 7 days. Table 3 shows the frequency of AEs in the 14 patients. Proteinuria, which was the most frequent factor for dose reduction or interruption, was observed in 85% (11/13) of patients in any grade including 23% (3/13) in grade 3. Other frequent AEs were malaise in 71% (10/14) of patients in any grade, hypothyroidism in 57% (8/14) of patients in any grade, and hypertension in 50% (7/14) of patients in any grade. Additionally, the increase in serum creatine kinase in any grade was observed in 21% (3/14) of patients without myalgia and muscle cramp.

### 2.3. Pharmacokinetic Analysis

Results of the pharmacokinetic analysis of plasma concentrations of cabozantinib over 6 weeks after the start of administration are shown in Figure 1. Maximum trough concentration (C_trough_) was achieved until a few weeks after the initial administration, and C_trough_ decreased over time in accordance with dose modifications because of AEs. Six patients discontinued cabozantinib during the 6-week period because of proteinuria, malaise, severe diarrhea, or sudden death. C_trough_ at 6 weeks after excluding two outliers was appropriately 500 ng/mL. There was no significant correlation between the starting dose and maximum C_trough_ (*r* = 0.380, *p* = 0.180) (Appendix A). As shown in Table 4, the mean value of median trough concentration (C_trough median_) and dose-normalized C_trough median_ were 755.6 ± 556.1 ng/mL and 23.8 ± 16.9 ng/mL/mg, respectively. The median value of C_trough median_ was 653.5 ng/mL. The coefficient of variation (CV) of C_trough median_ and dose-normalized C_trough median_ was 73.6% and 71.0%, respectively. The mean values of maximum C_trough_ and dose-normalized maximum C_trough_ were 996.0 ± 611.4 ng/mL and 31.6 ± 18.7 ng/mL/mg, respectively. The CV of maximum C_trough_ and dose-normalized maximum C_trough_ was 61.4% and 59.2%, respectively. 

### 2.4. Changes in Urine Protein and Muscle Enzymes

The time courses of urine protein-to-creatinine ratio (UPCR), serum creatine kinase, and serum aldolase are shown in Figure 2. The UPCR in 13 patients treated with cabozantinib (excluding the dialysis patient), at 1, 2, 4, and 6 weeks, was significantly higher than baseline UPCR values (*p* = 0.016, *p* = 0.003, *p* = 0.028, and *p* = 0.037, respectively). Serum creatine kinase and aldolase values at 1, 2, 3, 4, and 6 weeks were significantly higher than baseline values (creatine kinase; *p* = 0.005, *p* = 0.002, *p* = 0.007, *p* = 0.002, and *p* = 0.007, aldolase; *p* = 0.002, *p* < 0.001, *p* = 0.008, *p* = 0.005, and *p* = 0.008, respectively). Additionally, the proportion of patients who exceeded standard values in serum creatine kinase and aldolase were 21.4% and 100%, respectively.

### 2.5. Associations between Changes in Urine Protein or Muscle Enzymes and Cabozantinib Exposure 

Associations between changes from baseline in UPCR, serum creatine kinase, and aldolase and C_trough_ of cabozantinib at any point during the 6-week period are shown in Figure 3A–C. There were significant positive correlations between changes in UPCR, serum creatine kinase, and aldolase values at 1, 2, 3, 4, and 6 weeks and each C_trough_ of cabozantinib at the same time points (*r* = 0.348, *p* = 0.012; *r* = 0.581, *p* < 0.001; *r* = 0.664, *p* < 0.001, respectively). Although there was no significant association between maximum changes in UPCR and maximum C_trough_ (*r* = 0.368, *p* = 0.216) (Figure 3D), highly significant statistical correlations were observed between both maximum changes in serum creatine kinase and aldolase values during treatment and maximum C_trough_ (*r* = 0.736, *p* < 0.01; *r* = 0.798, *p* < 0.001, respectively) (Figure 3E,F). 

### 2.6. Receiver Operating Characteristic Curve Analysis

The median value of maximum (929.0 ng/mL) was used to divide patients into high-maximum C_trough_ (≥929.0 ng/mL) and low-maximum C_trough_ (<929.0 ng/mL) groups. As shown in Table 5, a receiver operating characteristic (ROC) curve analysis was performed and all areas under the ROC (AUROC) were calculated, including high-maximum C_trough_ at each point during cabozantinib treatment. The ROC analysis revealed that changes from baseline in UPCR ≥0.0911 g/gCr, serum creatine kinase ≥ 70.5 U/L, and aldolase ≥ 6.1 U/L at any point during treatment were the ideal cutoff values for predicting inclusion of high-maximum C_trough_. Moreover, both changes in creatine kinase and aldolase were relatively highly accurate in predicting high-maximum C_trough_ with AUROC of 0.929 for creatine kinase and AUROC of 0.833 for aldolase.

### 2.7. Therapeutic Effect and Cabozantinib Exposure

According to the assessment of therapeutic response (excluding one patient without computed tomography (CT) assessment), partial response, stable disease, and progressive disease were observed in one, 10, and two patients, respectively. The best objective response rate and disease control rate was 7.7% and 84.6%, respectively. The median progression-free survival (PFS) for all patients treated with cabozantinib was 87 d (95% confidence interval (CI): 74.2–99.8 d) (Figure 4A). The median PFS for patients with high-C_trough median_ and low-C_trough median_ was 87 d (95% CI: 76.7–97.3 d) and 143 d (95% CI: 0.0–309.8 d), respectively (Figure 4B). There was no significant difference between the two groups (*p* = 0.432).

## 3. Discussion

The present study showed that cabozantinib for treatment of uHCC was associated with a high frequency of drug interruption because of AEs, and that there was large interindividual pharmacokinetic variability among uHCC patients. Additionally, the increases in serum creatine kinase and aldolase levels were strongly associated with cabozantinib exposure.

Contrary to former reports, including the international Phase III trial and real-world data [3,5,6,7], our cohort had a 100% rate of dose reduction, despite the initial dose being less than 60 mg in most cases. Notably, our pharmacokinetic analysis showed heterogeneity in the CV values from exposure to cabozantinib (60–70%). Therefore, it was difficult to determine an optimal dose of cabozantinib with good balance of efficacy and toxicity. Even though cabozantinib exposure ≥ 750 ng/mL was proposed to be a target value for the optimal treatment outcome in renal cell carcinoma [8], targeted exposure is considered to vary with tumor types [9]. In the exposure–response models of the CELESTIAL trial, predictive average cabozantinib concentrations of 20, 40, and 60 mg were 383, 766, and 1148 ng/mL, respectively [10]. However, the C_trough median_ and focused C_trough_ values at 6 weeks in our cohort were 653.5 ng/mL and appropriately 500 ng/mL, respectively. Although the starting dose of 60 mg cabozantinib was predicted to induce a good therapeutic response compared with that of 40 or 20 mg [11], the initial dose of 60 mg, at least in Japanese patients, may need further consideration. The contributing factor of the higher toxicity in Japanese patients during cabozantinib treatment than that in Western patients may be a difference in physique. Kanzaki et al. reported that the nephron number in Japanese patients with hypertension or chronic kidney disease is lower than that in other races [12]. Renal sensitivity to anti-VEGF therapy may be also different between Western and Japanese patients. Cabozantinib has a long terminal plasma half-life of approximately 120 h and its area under blood concentration time curve increased five-fold by day 15 of daily dosing with 60 mg [13], indicating that a feature of the drug is high accumulation. Therefore, caution is required when making judgments on whether to continue or interrupt the drug.

Our cohort, which included 43% with past history of atezolizumab plus bevacizumab and 93% with that of lenvatinib, reflects current clinical practice in that atezolizumab plus bevacizumab is used as first-line treatment and lenvatinib is often used as second-line treatment for uHCC [14]. The difference in median PFS at 3.6 months in the sorafenib-naïve cohort from the Japanese Phase II trial [4] and that at 2.9 months in our cohort can be ascribed to differences in past history of systemic therapy.

A recent pharmacokinetic study of lenvatinib in HCC patients showed that the CV value of C_trough_ was 57% [15]. Likewise, our study indicated that the CV value of C_trough median_ and maximum C_trough_ dose-normalized to 20 mg was approximately 70% and 60%, respectively. The large pharmacokinetic variability of cabozantinib can therefore impact its safety and efficacy during therapy for uHCC. Theoretically, compared with dose–response, exposure-response can be more useful for investigating the effect of drugs on tumor response and AEs [16]. An exposure-response would be especially applicable for treatment of HCC using cabozantinib because of the large pharmacological variability.

In the Japanese phase II trial, proteinuria was observed in 29% of patients in the sorafenib-naïve cohort, with median age of 73 years and without past history of atezolizumab plus bevacizumab [4]. In contrast, 85% of patients in our cohort, with a median age of 80 years, developed proteinuria in any CTCAE grade including 23% in grade 3. This may be explained by differences in patient features, such as age and past history of anti-VEGF therapy. Patients on regimens containing bevacizumab or lenvatinib often developed proteinuria by inhibition of VEGF [17,18]. Proteinuria, which potentially precludes long-term administration of TKIs, is associated with exposure to TKIs targeting VEGF signaling [19,20]. In the clinical setting, because cabozantinib is administered as later-line systemic chemotherapy for uHCC, it is assumed that pre-existing TKI-induced renal stress can enhance the development of proteinuria. Previous in vitro data on the half-maximal inhibitory concentration of TKIs indicated that cabozantinib had greater affinity for VEGFR-2 compared with other TKIs such as lenvatinib and sorafenib [21]. Inadequate interruption or dose reduction of cabozantinib could result in severe and long-term renal damage. Taken together, optimal setting of the starting dose and dose modification of cabozantinib, in addition to consideration of drug exposure, would be preferable. Krens et al. proposed a treatment algorithm on the basis of measured cabozantinib exposure at 4 and 12 weeks after starting treatment and every 24 weeks thereafter, and the clinical response in patients with renal cell carcinoma [8]. However, plasma monitoring of cabozantinib is not necessarily acceptable in the clinical setting.

Although they receive less attention, we analyzed the time course of muscle enzymes, including serum aldolase. Aldolase, a glycolytic enzyme that is widely distributed in the living body, is often measured together with serum creatine kinase in cases of muscle injury [22]. We found that serum creatine kinase was elevated even within normal range during cabozantinib treatment. Interestingly, in all patients, serum aldolase levels were also elevated, and exceeded standard values. To our knowledge, there are few comprehensive reports in the literature on the time course of muscle enzymes during cabozantinib treatment.

It was reported that the increase in serum creatine kinase correlated with the dose of imatinib in patients with chronic myeloid leukemia [23]. Notably, we showed that both serum creatine kinase and aldolase values were associated with cabozantinib exposure. Additionally, maximum changes in serum creatine kinase and aldolase values were highly accurate for predicting whether patients were administered a sufficient drug concentration. Clinicians must pay attention to changes in the levels of muscle enzymes, which may represent an indicator of cabozantinib exposure. For example, in cases of cabozantinib dose escalation, knowledge of the levels of serum creatine kinase and aldolase would be useful.

A recent in vitro and animal study found that lenvatinib induced muscle injury through the reduction in various mitochondrial proteins in skeletal muscle [24]. Tyrosine kinase receptors can activate the phosphoinositide 3-kinase (PI3K)/thymoma viral proto-oncogene (AKT)/mechanistic target of rapamycin kinase (mTOR) pathway, an important intracellular signaling cascade for protein synthesis [25]. In addition to lenvatinib, cabozantinib may cause muscle injury by suppressing PI3K/AKT/mTOR signaling.

Our study had some limitations. First, this was a single-institution study on a limited number of patients. ROC analysis had weak applicability because of the small sample size. Therefore, an exposure-efficacy association during cabozantinib treatment could not be identified. A larger prospective study is required to investigate the relationship between cabozantinib exposure and the therapeutic response including the effect of previous treatment. Nonetheless, the changes in muscle enzymes were highly associated with cabozantinib exposure, despite the small cohort. Second, measurements in the pharmacokinetic analysis were taken exclusively at C_trough_. Despite these limitations, monitoring muscle enzymes may be informative for enhancing pharmacological therapy with cabozantinib in uHCC patients.

In conclusion, our findings suggest that there are large interindividual differences in cabozantinib exposure among Japanese patients with uHCC. Measurement of muscle enzymes such as creatine kinase and aldolase might enhance the clinical usefulness of cabozantinib therapy.

## 4. Materials and Methods

### 4.1. Patients

We performed a single-center, prospective, observational study in uHCC patients treated with cabozantinib. Japanese patients who were administered cabozantinib as a subsequent therapy following failure of other systemic therapies between March 2021 and March 2022 at Juntendo University Nerima Hospital were enrolled in the study. Written informed consent was obtained from all patients for participation in the study. The study was approved by the Ethical Committee of our university (N20-0072) and was performed in accordance with the ethical standards of the 1964 Declaration of Helsinki and its later amendments. The study was registered with the UMIN Clinical Trials Registry as UMIN000043775. Patients with impaired Eastern Cooperative Oncology Group performance status (PS 2, 3, and 4) were excluded from the study. Patients took oral cabozantinib (CABOMETYX; Exelixis, Inc. Alameda, CA, USA) at 60, 40, or 20 mg once daily or 20 mg on alternate days in the fasted state (≥2 h after breakfast, with no food ingested for ≥1 h after dosing) as an initial dose according to the discretion of attending physicians. Body weight, liver impairment, and toxicity of previous treatment were also taken into account. Blood and urine samples were obtained from all patients to observe AEs on at minimum days 7, 14, 21, 28, and 42. Serum creatine kinase and aldolase values were examined using the Japan Society of Clinical Chemistry transferable method and ultraviolet enzyme method, respectively. AEs were assessed according to the Common Terminology Criteria for Adverse Events (CTCAE) version 5.0. According to a guide for cabozantinib administration, the dose was reduced upon development of any CTCAE grade 3 or 4, or any unacceptable grade 2. If tolerated, the dose was elevated. Therapeutic response was assessed using enhanced CT or plain CT in patients with renal impairment, and images were captured every 6 (+2) weeks following the start of treatment in accordance with Response Evaluation Criteria in Solid Tumors (RECIST) guidelines version 1.1.

### 4.2. Pharmacokinetic Analysis of Cabozantinib

C_trough_ of cabozantinib were measured in plasma samples collected at 7, 14, 21, 28, (35), and 42 d, up to 6 weeks or discontinuation of cabozantinib. The timing of measurement was 22–26 h after the final dosing of the prior day. Furthermore, C_trough median_ of cabozantinib was calculated for each C_trough_ at 7, 14, 21, 28, (35), and 42 d. After plasma samples were isolated by centrifugation within 30 min at 1500 rpm for 10 min at 4 °C, they were deep-frozen at −80 °C until measurement. Plasma cabozantinib concentrations were measured using liquid chromatography-tandem mass spectrometry. The lower limit of quantitation for cabozantinib in plasma was 10 ng/mL. Assay performance was monitored using quality control (QC) samples at concentrations of 30, 300, and 8000 ng/mL. The accuracy of QC samples ranged from 101.0–111.4%.

### 4.3. Statistical Analysis

Continuous variables are presented as median (range). The Wilcoxon signed-rank test was used for comparisons of UPCR, serum aldolase, and serum creatine kinase between baseline values and values during treatment. Spearman’s correlation coefficient was used to determine the associations between pairs of variables. ROC curve analysis was performed to identify the cutoff value of changes from baseline in UPCR, creatine kinase, and serum aldolase for predicting which patients will be included in the high cabozantinib exposure group. AUROCs are presented with 95% CIs. PFS was calculated using the Kaplan–Meier method. The log-rank test was used for comparisons between two groups. All tests were two-sided, and *p* values < 0.05 were considered statistically significant. Statistical analyses were performed using SPSS Statistics for Windows, Version 27 (IBM Corp., Tokyo, Japan).

## Figures and Tables

**Figure 1 pharmaceuticals-15-01460-f001:**
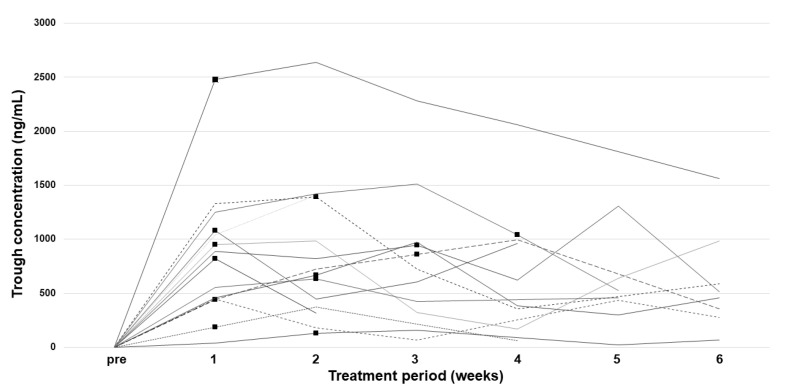
Time course of plasma trough concentrations of cabozantinib during treatment. Black squares indicate the point of the first dose reduction or interruption.

**Figure 2 pharmaceuticals-15-01460-f002:**
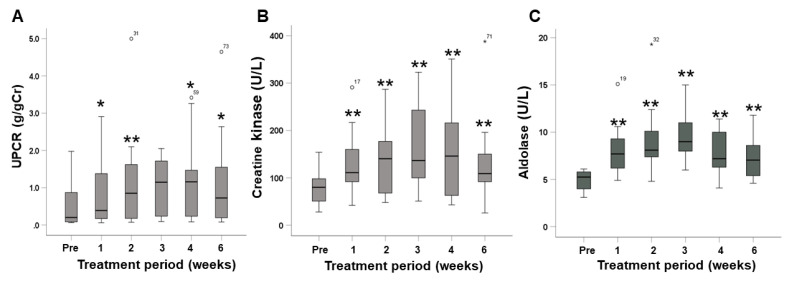
Time courses of the urine protein-to-creatinine ratio (UPCR) (**A**), serum creatine kinase (**B**), and serum aldolase (**C**) during cabozantinib treatment using box-and-whisker plots. The boxes span data between two quartiles (IQR, interquartile range), while the central line in each box represents the median of each group. The vertical lines represent the smallest and largest values that are not outliers. Outliers are values between 1.5 and 3 IQRs from the end of the box. * *p* < 0.05 vs. pre. ** *p* < 0.01 vs. pre.

**Figure 3 pharmaceuticals-15-01460-f003:**
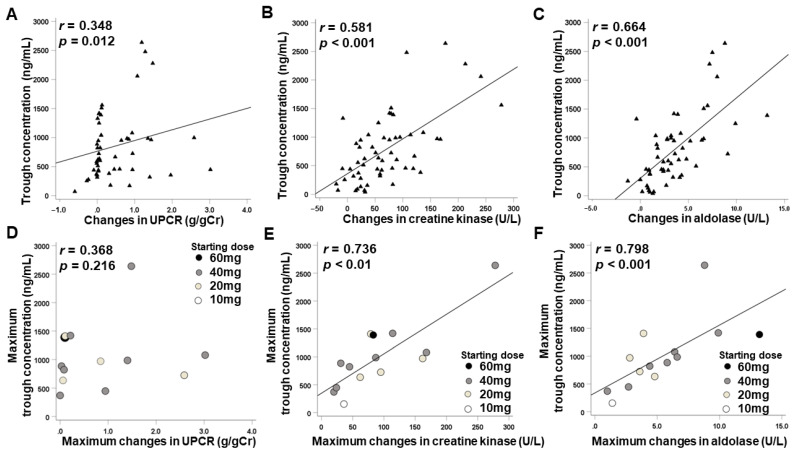
Correlations between all values of changes from baseline in urine protein-to-creatinine ratio (UPCR), serum creatine kinase, and serum aldolase and C_trough_ of cabozantinib at any point (**A**–**C**), and correlations between maximum changes in UPCR, serum creatine kinase, and serum aldolase and maximum trough concentration of cabozantinib (**D**–**F**).

**Figure 4 pharmaceuticals-15-01460-f004:**
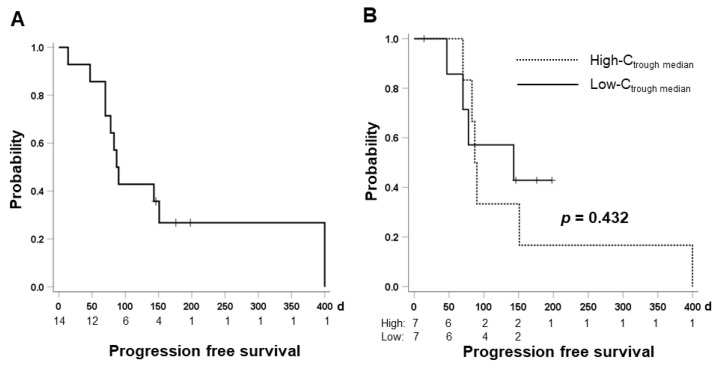
Progression-free survival (PFS) for all patients treated with cabozantinib (**A**) and comparison of the PFS between patients included in the high-C_trough median_ (≥653.5 ng/mL) and low-C_trough median_ (<653.5 ng/mL) groups (**B**).

**Table 1 pharmaceuticals-15-01460-t001:** Patient features.

□	N = 14
Age, years	80 (69–91)
Sex, male/female	13/1
Etiology: HCV/NASH/ALC/PBC	5/5/3/1
Performance status ECOG: 0/1	12/2
Body weight, kg	58 (47–70)
Body surface area, m^2^	1.66 (1.45–1.77)
Body mass index, kg/m^2^	22.50 (17.26–26.56)
BCLC Stage: B/C	6/8
Treatment line: 2nd/3rd/5th	6/7/1
Past history of ATZ + BEV: Yes/No	6/8
Past history of lenvatinib: Yes/No	13/1
Child-Pugh score: 5/6/7/8	6/6/1/1
mALBI grade: 1/2a/2b	2/8/4
Starting dose: 60/40/20/10 mg	1/8/4/1
Total bilirubin, mg/dL	0.8 (0.1–2.3)
Albumin, g/dL	3.6 (3.1–3.4)
Prothrombin activity, %	80 (61–126)
eGFR, mL/dL/1.13 m^2^	54.33 (7.08–87.76)
Urinary protein-to-creatinine ratio ^#^, g/gCr	0.202 (0.06–1.98)
Creatine kinase, U/L	80 (28–154)
Aldolase, U/L	5.25 (3.1–6.5)
AFP, ng/mL	22.5 (1–20,200)
DCP, mAU/mL	9780 (20–78,100)

Median (range) or n; ^#^: excluding the one dialysis patient. AFP, α-fetoprotein; ALC, alcohol; ATZ, atezolizumab; BCLC, Barcelona Clinic Liver Cancer; BEV, bevacizumab; eGFR, estimated glomerular filtration rate; HBV, hepatitis B virus; HCV, hepatitis C virus; DCP, des-γ-carboxy prothrombin; mALBI, modified albumin-bilirubin; NASH, nonalcoholic steatohepatitis; PBC, primary biliary cholangitis; PS, performance status.

**Table 2 pharmaceuticals-15-01460-t002:** Therapeutic details of cabozantinib administration.

□	Total (N = 14)
Median duration of drug exposure, days (range)	70 (7–380)
Median 2-week relative dose intensity, % (range)	33.3 (16.7–100)
Median 4-week relative dose intensity, % (range)	30.6 (12.5–66.7)
Dose reduction rate, %	100
Median duration of dose reduction, days (range) *	14 (7–28)
Dose interruption rate, %	79
Median duration of dose interruption, days (range) ^§^	7 (7–70)

* n = 14 in the total population, ^§^ n = 11 in the total population.

**Table 3 pharmaceuticals-15-01460-t003:** Adverse events in any grade during cabozantinib treatment.

□	Total	Grade 1	Grade 2	Grade 3	Any grade (%)	Grade 3 (%)
Proteinuria	13	3	5	3	85	23
Malaise	14	3	5	2	71	14
Anorexia	14	4	2	0	42	0
Hand and foot syndrome	14	3	1	0	28	0
Hypothyroidism	14	2	6	0	57	0
Diarrhea	14	1	0	2	21	14
Hypertension	14	5	2	0	50	0
Increased AST/ALT	14	1	1	1	21	7
Increased CK	14	1	2	0	21	0
Decreased PLT	14	0	2	0	14	0

ALT, alanine aminotransferase; AST, aspartate aminotransferase; CK, creatine kinase; PLT, platelet.

**Table 4 pharmaceuticals-15-01460-t004:** Pharmacokinetic parameters during cabozantinib therapy.

C_trough median_	Mean ± SD	755.6 ± 556.1
	median (range), ng/mL	653.5 (68.2–2280.0)
	CV, %	73.6
Dose-normalized C_trough median_	mean ± SD	23.8 ±16.9
	median (range), ng/mL/mg	20.6 (5.0–6.1)
	CV, %	71.0
Maximum C_trough_	mean ± SD	996.0 ± 611.4
	median (range), ng/mL	929.0 (155.0–2640.0)
	CV, %	61.4
Dose-normalized maximum C_trough_	mean ± SD	31.6 ± 18.7
	median (range), ng/mL/mg	25.8 (9.3–70.5)
	CV, %	59.2

C_trough_, trough concentration; CV, coefficient of variation; SD, standard deviation. Dose-normalized data represent the concentration/dose of cabozantinib.

**Table 5 pharmaceuticals-15-01460-t005:** ROC curve analysis of each parameter and prediction of high-maximum C_trough_ (≥929.0 ng/mL).

	Cutoff Value	AUROC (95% CI)	*p* Value	Sensitivity, %	Specificity, %	PPV, %	NPV, %
Maximum changes in UPCR (g/gCr)	0.0911	0.762 (0.463–1.000)	0.116	100.0	66.7	50.0	60.0
Maximum changes in creatine kinase (mg/dL)	70.5	0.929 (0.776–1.000)	0.010	100.0	83.3	100.0	87.5
Maximum changes in aldolase (mg/dL)	6.1	0.833 (0.600–1.000)	0.046	66.7	82.1	77.8	100.0

AUROC, area under the receiver operating characteristic curve; CI, confidence interval; C_trough median_, median trough concentration, NPV, negative predictive value; PPV, positive predictive value; UPCR, urinary protein-to-creatinine ratio.

## Data Availability

Data is contained within the article and Appendix A.

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
