# Peer review of "Impact of Cabozantinib Exposure on Proteinuria and Muscle Toxicity in Patients with Unresectable Hepatocellular Carcinoma"

_pharmaceuticals, 2022, doi:10.3390/ph15121460_

Round 1

Reviewer 1 Report

Overall, this reviewer believes that the analyses being performed here are being grouped together inappropriately and runs the serious risk of biasing the relationship observed.

General comments:

** Discussion should include some clinical thoughts on 1) why/what factors contribute to Japanese patients experiencing higher toxicity (beyond saying higher plasma levels) and 2) what is the clinical decision making occurring for the dose reduction for initiation of therapy.

** There is exposure-response and response-toxicity analysis data included in RCC and HCC from a few papers and you can query the EMA and FDA drug approval Clinical Pharmacology review documents. These data and knowledge around how a target concentration had been identified, but studies in real-world patients seem to be lower should be included to provide context. It will also provide a context of your concentration cut-off level in comparison to prior literature.

Sample literature:

https://bmccancer.biomedcentral.com/articles/10.1186/s12885-022-09338-1#:~:text=For%20the%2040%20mg%20dose,had%20increased%20to%201.4%2Dfold.

https://www.ncbi.nlm.nih.gov/pmc/articles/PMC5973957/

https://www.ncbi.nlm.nih.gov/pmc/articles/PMC9291492/

** The separation of Ctrough values into high and low cohorts based on all data without taking into account treatment week is flawed. You note how accumulation occurs, which even gives rationale for how levels in week 1 & 2 are not directly comparable for later weeks. This can also results in a false association between treatment week groups to be formed, and generates a relative association of ~0.6.

** The ROC analysis is fair to poor, especially when we consider the confidence intervals. Given the small sample size, the discussion should include how ROC may not be appropriate and more data are necessary. Given the current results, this reviewer thinks you are overstating your results.

** Reanalysis at 1) starting dose; 2) common dose (like 40mg or 20 mg) at initiation + mid-therapy (Wk 3 or 4); and 3) best-tolerated dose [BTD] would be a significant improvement over the current mass pooling of data.

** Did the authors ever analyze that, at BTD, is the Ctrough concentration similar among the doses administered?

** Cabozantinib dose reductions, a surrogate for toxicity and adequate drug exposure, appear to be associated with improved TTF and OS in mRCC. Toxicity driven/individualized dosing strategies for cabo alone and in combination with immunotherapy, warrant further investigation. You can make some bridges into HCC.

Specific comments are below:

** Line 24-25: “…relatively accurately predicts…”  This is not definitive and sounds odd. It almost sounds like you ‘think’ it is accurate but can’t say it actually is.

** Never clarified, but were all subjects on monotherapy? If not this needs to be considered in the analysis. Realistically, so would disease progression/tumor size, but this reviewer understands the limited sample size.

** Figure 1: It would be useful to note where on lines where a treatment decrease occurred (just add a symbol on the line for the first measurement after a dose reduction).

** Table 4: The use of “Ctrough median, dose normalized to 20mg” is confusing as classical PK parameter tables would suggest you are presenting dose-normalized data (i.e., CONC / DOSE). Other comments above on suggested reanalysis would likely remove this parameter.

** Figure 3: As this study is ultimately hypothesis-generating, visualization of dose as the symbol can allow to see if clustering/association occurs. 

** The review of compliance to therapy is not noted. Being a prospective study and having a direct effect on PK, these data should be included and subjects with compliance issues noted.

Author Response

Responses to the reviewers’ comments

#1. Reviewer 1

General comments:

** Discussion should include some clinical thoughts on 1) why/what factors contribute to Japanese patients experiencing higher toxicity (beyond saying higher plasma levels) and 2) what is the clinical decision making occurring for the dose reduction for initiation of therapy.

We thank the reviewer for the clinically important comment.

1) The contributing factor of the higher toxicity in Japanese patients during cabozantinib treatment than that in Western patients may be a difference in physique. Kanzaki et al. reported that the nephron number in Japanese patients with hypertension or chronic kidney disease is lower than that in other races. Renal sensitivity to anti-VEGF therapy may be also different between Western and Japanese patients. We have added this information to the revised manuscript (lines 201–206).

2) The initial dose was decided by attending physicians while taking into consideration body weight, liver impairment, and toxicity of the previous treatment. We have added this information to the revised manuscript (lines 296–297).

** There is exposure-response and response-toxicity analysis data included in RCC and HCC from a few papers and you can query the EMA and FDA drug approval Clinical Pharmacology review documents. These data and knowledge around how a target concentration had been identified, but studies in real-world patients seem to be lower should be included to provide context. It will also provide a context of your concentration cut-off level in comparison to prior literature.

Sample literature:

https://bmccancer.biomedcentral.com/articles/10.1186/s12885-022-09338-1#:~:text=For%20the%2040%20mg%20dose,had%20increased%20to%201.4%2Dfold.

https://www.ncbi.nlm.nih.gov/pmc/articles/PMC5973957/

https://www.ncbi.nlm.nih.gov/pmc/articles/PMC9291492/

We appreciate the reviewer’s useful comment. We have described the focused value of Ctrough at 6 weeks in the revised manuscript. Subsequently, discussion relating to the target concentration including different tumor types has been added to the revised manuscript (lines 193–198), and appropriate references have been added.

** The separation of Ctrough values into high and low cohorts based on all data without taking into account treatment week is flawed. You note how accumulation occurs, which even gives rationale for how levels in week 1 & 2 are not directly comparable for later weeks. This can also results in a false association between treatment week groups to be formed, and generates a relative association of ~0.6.

We appreciate the reviewer for pointing out this issue. Taking into consideration a high accumulation of cabozantinib, we recalculated the ROC curve on the basis of the median value of maximum Ctrough. As expected, we identified an even higher accuracy rate for changes in creatine kinase and aldolase concentrations for predicting inclusion in the high maximum Ctrough group. Our results have been added to the revised version of Table 5 and explained in lines 151–160 in the text.

** The ROC analysis is fair to poor, especially when we consider the confidence intervals. Given the small sample size, the discussion should include how ROC may not be appropriate and more data are necessary. Given the current results, this reviewer thinks you are overstating your results.

In accordance with the reviewer’s comment, we have mentioned that ROC curve analysis may have had have weak applicability owing to the small sample size in the limitations part of the manuscript. Moreover, we have toned down our conclusions of our results in the revised manuscript.

** Reanalysis at 1) starting dose; 2) common dose (like 40mg or 20 mg) at initiation + mid-therapy (Wk 3 or 4); and 3) best-tolerated dose [BTD] would be a significant improvement over the current mass pooling of data.

We thank the reviewer for this suggestion. We reanalyzed the correlation between the starting dose and maximum Ctrough. Consequently, we identified no significant correlation between the starting dose and maximum Ctrough(r = 0.380, p = 0.180) (see Supplementary Figure 1 in the revised manuscript). We agree that an analysis including a common dose at mid-therapy (week 3 or 4) and the best-tolerated dose would be useful. However, as shown in Figure 1 in the original manuscript, a high frequency of dose reduction and/or interruption occurred even during the initial 6 weeks, and the completion rate at week 6 of cabozantinib treatment was 57% (8/14). In addition, dose modification was performed even after 6 weeks on many occasions. Therefore, an analysis of a common dose or the best tolerated dose is not possible in our small cohort.

** Did the authors ever analyze that, at BTD, is the Ctrough concentration similar among the doses administered?

Because an analysis of the best tolerated dose was infeasible, we reanalyzed the correlation between the starting dose and maximum Ctrough (see Supplementary Figure 1 in the revised manuscript).

** Cabozantinib dose reductions, a surrogate for toxicity and adequate drug exposure, appear to be associated with improved TTF and OS in mRCC. Toxicity driven/individualized dosing strategies for cabo alone and in combination with immunotherapy, warrant further investigation. You can make some bridges into HCC.

We thank the reviewer for the insightful comment. We have discussed the possibility of a treatment algorithm based on cabozantinib exposure and the clinical response and cited a recent report by Krens et al. in the revised manuscript (lines 240–243).

Specific comments are below:

** Line 24-25: “…relatively accurately predicts…”  This is not definitive and sounds odd. It almost sounds like you ‘think’ it is accurate but can’t say it actually is.

In accordance with the reviewer’s comment, we have deleted “relatively.”

** Never clarified, but were all subjects on monotherapy? If not this needs to be considered in the analysis. Realistically, so would disease progression/tumor size, but this reviewer understands the limited sample size.

In unresectable hepatocellular carcinoma, cabozantinib can be administered as a monotherapy following the use of systemic therapies. As pointed out by the reviewer, we have referred to the necessity for further study to investigate cabozantinib exposure and the therapeutic response in the limitations part of the revised manuscript.

** Figure 1: It would be useful to note where on lines where a treatment decrease occurred (just add a symbol on the line for the first measurement after a dose reduction).

We thank the reviewer for this comment about improving the quality of original Figure 1. We have added black squares indicating the point of the first dose reduction or interruption to this figure.

** Table 4: The use of “Ctrough median, dose normalized to 20mg” is confusing as classical PK parameter tables would suggest you are presenting dose-normalized data (i.e., CONC / DOSE). Other comments above on suggested reanalysis would likely remove this parameter.

We recalculated the dose-normalized data representing concentration/dose of cabozantinib and have shown the results in Table 4 of the revised manuscript.

** Figure 3: As this study is ultimately hypothesis-generating, visualization of dose as the symbol can allow to see if clustering/association occurs.

Cabozantinib has a pharmacological feature of high accumulation. Therefore, illustrating the relationship between Ctrough at any point during the 6-week period and changes in toxicities according to the administrated dose at any point is not necessarily meaningful. However, for considering relationships between maximum Ctrough and maximum changes in renal and muscle toxicities, visualization based on the starting dose would be convenient for readers of the Journal. Therefore, we have changed the triangles in Figure 3D–F of the original manuscript to open-colored circles to indicate the starting dose in the revised manuscript.

** The review of compliance to therapy is not noted. Being a prospective study and having a direct effect on PK, these data should be included and subjects with compliance issues noted.

In accordance with the reviewer’s comment, we have added information about medication compliance to the revised manuscript (lines 74–75).

Reviewer 2 Report

The authors have done a study with a small cohort of uHCC patients from a single institution and have shown a large pharmacological variability with cabozantinib exposure among Japanese patients and have demonstrated the adverse effects with this treatment, which led to dose reduction in all the patients. The authors have tried to highlight the significance of paying attention to cabozantinib exposure and using enzyme such as aldolase and creatine kinase as markers to monitor cabozantinib exposure to get a better anticipation of possible adverse effects and the need to reduce dosage.

I agree with the authors that there biggest limitation is the cohort sample size and they have clearly mentioned that in their text. However, they should also highlight that there is a lot of variability in that small group due to previous treatment differences. 

Author Response

Responses to the reviewers’ comments

#2. Reviewer 2

The authors have done a study with a small cohort of uHCC patients from a single institution and have shown a large pharmacological variability with cabozantinib exposure among Japanese patients and have demonstrated the adverse effects with this treatment, which led to dose reduction in all the patients. The authors have tried to highlight the significance of paying attention to cabozantinib exposure and using enzyme such as aldolase and creatine kinase as markers to monitor cabozantinib exposure to get a better anticipation of possible adverse effects and the need to reduce dosage.

I agree with the authors that there biggest limitation is the cohort sample size and they have clearly mentioned that in their text. However, they should also highlight that there is a lot of variability in that small group due to previous treatment differences.

We thank the reviewer for highlighting this problem. In accordance with the reviewer’s comment, we have added a comment in the limitations part of the revised manuscript (Lines 271–273).

Round 2

Reviewer 1 Report

Thank you for reviewing my prior comments and making the effort to revise your original manuscript.

 The re-analysis is at least more in line with sound designs, but the reviewer recognizes the small sample size, so the conclusions are hypothesis generating and overall conclusions of this manuscript reflect a less definitive answer with the need for further study. This is an appropriate outcome for this level of data.

Only minor edits would be:

** Line 84: This addition on compliance adds nothing to the methods or justification of results. "Adequate" does not denote a percentage either, and my guess is that this factor was not considered. It would be better to remove and note compliance was not tracked as a limitation.

** Line 264: “An exposure-response with an appropriate marker of response in HCC ..." may be more descriptive. I think the main goal of exposure (versus dose) response is on point, but identifying the appropriate marker to leverage exposures is also not fully supported in HCC; but this is part of what you are hypothesizing as well (in aldolase and/or creatine kinase).